# Peer review of "Expanded PCR Panel Testing for Identification of Respiratory Pathogens and Coinfections in Influenza-like Illness"

_diagnostics, 2023, doi:10.3390/diagnostics13122014_

Round 1

Reviewer 1 Report

The article entitled “Expanded PCR Panel Testing for Identification of Respiratory Pathogens and Coinfections in Influenza-like-Illness” is a research about the identification of the different pathogens such as bacteria, viruses and fungi in 381 nasopharyngeal and anterior nasal swabs  from individual of all ages with symptoms of acute respiratory tract infection (RTI). This clinical research is important as the respiratory tract infections are cause of major morbidity and mortality burdens worldwide.  In addition, the co-infections between bacteria and virus or bacteria and fungus are associated with severity of disease. Based on my knowledge, there are little studies that investigate all pathogens more prevalent and common in the RTI. The authors tested all samples using a multiplex expanded panel testing  (TAC, OA) for the presence of bacterial, viral and fungal respiratory pathogens and the identification of mono- poly-microbial infections. The results of this study show an incidence of co-infections of 40.36% and 36.58% for two or more organisms on TAC and OA. The types of prevalent pathogens were the viruses (60%) followed by bacteria (40%) and fungi (1%). I think that this study was well conducted in terms of project work and methods and the results are interesting. Therefore, I think that this article is suitable for publication in its current version.

Author Response

We would like to thank the reviewer for taking the time to assess our manuscript.

There is no comment to address.

Reviewer 2 Report

Simultaneous genetic testing for COVID-19 and influenza and two
comprehensive pathogen tests were conducted to detect pathogens in 381 nasopharyngeal and anterior nasal swab samples from patients with respiratory tract infections collected from three different locations in the United States from February 2021 to May 2021.
SARS-CoV-2 was detected in only 84 samples (22%), and all samples were negative for influenza A or influenza B in two influenza detection assays. On the other hand, at least one pathogen organism was detected in about 58% of the 381 samples.

Without significant research funding, it would not be possible to conduct such a study, and the results of the study would be clinically useful.

Minor remarks.

As stated by the author, one of the objectives of this study was to evaluate two different PCR techniques for expanded panel testing (lines 303-313). This point should be added to the summary.

The summary should be revised as it contains only background and objectives and lacks results.

Author Response

We would like to thank the reviewer for taking the time to assess our manuscript. We have revised the manuscript and addressed the comments below.

Point 1: As stated by the author, one of the objectives of this study was to evaluate two different PCR techniques for expanded panel testing (lines 303-313). This point should be added to the summary.

Thank you for pointing this out. We agree with this comment. Therefore, we have updated the Abstract (Page 1) to include the evaluation of the two platforms as an objective.

Point 2: The summary should be revised as it contains only background and objectives and lacks results.

Thank you for pointing this out. We agree with this comment. Therefore, we have updated the Abstract (Page 1) to include a summary of the results.

Please see the revised manuscript in attachment. All revisions are marked up using the track changes function for an easy review. We would like to thank the referee again for taking the time to review our manuscript.
